# Social Responsibility and Spiritual Intelligence: University Students’ Attitudes during COVID-19

**DOI:** 10.3390/ijerph191911911

**Published:** 2022-09-21

**Authors:** Pedro Severino-González, Victoria Toro-Lagos, Miguel A. Santinelli-Ramos, José Romero-Argueta, Giusseppe Sarmiento-Peralta, Ian S. Kinney, Reynier Ramírez-Molina, Francisco Villar-Olaeta

**Affiliations:** 1Departamento de Economía y Administración, Facultad de Ciencias Sociales y Económicas, Universidad Católica del Maule, Talca 3480094, Chile; 2Escuela de Ingeniería Comercial, Facultad de Ciencias Sociales y Económicas, Universidad Católica del Maule, Talca 3480094, Chile; 3Facultad de Responsabilidad Social, Universidad Anáhuac, Huixquilucan 52780, Mexico; 4Ministerio de Educación, Ciencia y Tecnología de El Salvador, San Francisco Gotera 3201, El Salvador; 5Facultad de Ciencias Sociales, Universidad Tecnológica de El Salvador, San Salvador 06006, El Salvador; 6Facultad de Ciencias y Humanidades, Universidad Gerardo Barrios, Usulután 0614, El Salvador; 7Departamento de Tecnología Médica, Universidad Nacional Mayor de San Marcos, Lima 1500, Peru; 8Department of Foreign Languages, Central Washington University, Ellensburg, WA 98926, USA; 9Departamento de Ciencias Empresariales, Universidad de la Costa, Barranquilla 080001, Colombia; 10Departamento de Sociología, Ciencia Política y Administración Pública, Universidad Católica de Temuco, Temuco 4780000, Chile

**Keywords:** social responsibility, spiritual intelligence, COVID-19, educational, university students

## Abstract

Human behavior during COVID-19 has led to the study of attitude and preferences among the population in different circumstances. In this sense, studying human behavior can contribute to creating policies for integral education, which should consider the convergence between social responsibility and spiritual intelligence. This can lead to the sensitization of practices and attitude modification within society. The purpose of our research was to explore the spiritual intelligence attitudes of university students from the perspective of social responsibility, considering the sociodemographic characteristics of the research subjects during the COVID-19 pandemic. Our research design is quantitative and sectional, due to the use of two quantitative scales. The participants were university students from a city located in south-central Chile. A total of 415 participations were collected, of which 362 applications were valid. Statistically significant differences were found according to gender and age. Women and the student cohort between 18 and 24 years of age placed more importance on spiritual necessities. We thus highlight the necessity to have adequate spaces for spiritual intelligence training given its links with socially responsible behavior and, finally, the development of explanatory studies to determine its causalities. In practice, these results contribute to designing an educational policy on the formation of integral spiritual intelligence for future professionals.

## 1. Introduction

A range of social, political, and epistemological crises have led to the installation of challenges in the educational system, which are a consequence of widespread distrust, hopelessness, and lack of probity, evidenced in acts of corruption, injustice, pollution, deprivation, desolation, and vandalism [1]. For this reason, higher education institutions should place a sense of urgency on the noble challenge linked to the installation of work skills and ethical competencies [2,3] which can lead to the manifestation of acts based on honesty, integrity, respect, responsibility, transparency, and loyalty [4,5]. These can be achieved by implementing educational policies to procure an integral formation. This formation should be based on the principles of social responsibility, which allows the promotion of empathetic, supportive, ethical, altruistic, and sustainable behaviors [6,7,8].

Social responsibility is linked to ethical behavior, due to the search for the benefit to each person and the development of society in general [9], which could improve quality of life because of a high social conscience [10]. In this scenario, higher education institutions should contribute to social welfare as a product of their duty to exist and, therefore, of their substantive functions: knowledge generation, integral professional training, and promoting cultural events [11,12]. In this context, the transversality of social responsibility in each of their university processes could demonstrate socially responsible educational decisions and attitudes [13,14], forging values such as empathy, solidarity, prosociality, and social justice. 

All of the above must lead to renewed practices, which can incorporate knowledge, habits, and skills through curricular and extracurricular educational spaces and environments, with values such as solidarity, equity, and empathy [15]. These elements can be linked to intellectual and integral development, which includes mind, body, and character [1,16], based on a holistic, complex, and human approach [17]. In this scenario, the importance of rehumanization is found in its bases, some of them being the dignity of people [18], co-responsibility, and mutual care, which contribute to the recovery of the sense and meaning of life, where different ways of being and knowing are found in the path of coexistence [19].

The well-being and care of life in the times of the COVID-19 pandemic have become more relevant due to situations that have shown the fragility of the human being [20,21]. This has led international organizations and nations from different latitudes to implement strategies that seek to mitigate contagion by contact [22,23]. These strategies consider practices that range from the installation of a standard of conduct to formal and informal education [24,25]. Overall, these factors have highlighted the importance of comprehensive education, which must be timely, contextualized, sustainable and socially responsible [26,27].

The implementation of strategies that seek integral development of human beings can consider the sociodemographic characteristics of the wider population for research subjects [7,28]. This has recently motivated the study of spiritual intelligence in relation to certain sociodemographic characteristics of participants [29]. In this vein, we can consider the work of Parattukudi et al. [30] and Becerra and Becerra [31], which include, for example, gender, education, and age. There is also research that considers university students’ sociodemographic characteristics, such as age, gender, and marital status [32,33,34]. This does not leave aside cultural and ideological elements [35]. All these elements install challenges for the formation of spiritual intelligence as an HEI social responsibility strategy.

### 1.1. Social Responsibility and Spiritual Intelligence

Comprehensive training is a challenge that implies the provision of strategies which must respond to needs, according to the diverse characteristics and multiple aspects that comprise each human being. Future professionals’ attitudes must be competent and socially responsible [36] and sensitive to the challenges of modern society, as a result of a high social awareness in relation to the problems that afflict citizenship in general [37]. Values such as freedom, spirituality, and creativity are essential for holistic, transversal, and systematic understanding and teaching [38]. In this sense, the importance of education from a holistic approach contributes to the effectiveness of professional practice, which is stressed and enriched by spiritual capacities and needs, providing understanding, care, and compassion for oneself and others [39]. The orientations of higher education institutions should thus be comprehensive and consistent with the demands of the various territories, including public and social welfare, and operate at the physical, relational, psychological, and spiritual levels [2]. In line with this last concept, spiritual intelligence is an underlying concept. It is defined as the human ability to generate questions about the meaning of life and, understanding the meaning of them, simultaneously experiencing connections between each of us and the wider world that deliver expectations and transcendence [40].

Education in spiritual intelligence is linked with social responsibility, social awareness, and social sensitivity, due to the human need to be part of social, emotional, and family issues as a social being [41,42]. This is intended to contribute to harmonious, coherent, and consistent development in all areas of the human being [43]. The aforementioned elements help to install devices prevent incorrect, perverse, harmful and immoral attitudes, since a spiritually intelligent person is happy to do good and enjoys helping others [44]. Spiritual intelligence is difficult to achieve, since spirituality is an inherent component of individuals and complex to understand [45]. However, some approaches allow us to point out that spirituality can be understood from the meanings that people give to the relationship with God or a higher being, the links and union with nature, and the connections with others and, of course, at all times, with oneself by configuring and characterizing the self [46]. This facilitates the discovery of the role that an individual has in society and his or her contributions to community development, for which social awareness and self-knowledge are fundamental. This implies first discovering who we are and what we do, contributing to the development of spiritual intelligence [47], which provides incentives to face the difficulties of life and illuminates the answers to the fundamental questions of being [28]. 

Spiritual intelligence should thus be one of the functions that higher education institutions should develop, due to the contributions that could be generated to integral formation, providing competencies linked to critical thinking, self-knowledge, empathy, and ethical discernment [48], motivating the efforts of all the actors that make up an educational community and, especially, teachers. Teachers are mainly responsible for generating educational environments that allow the installation of moral and social values through reflective strategies [49]. These are all a product of spiritual intelligence when it is considered as the basis of cognitive intelligence [43]. In this sense, according to Sadiku and Musa [50], spiritual intelligence is one of the human abilities which involves a high individual and social awareness. It is also a consequence of a significant ability to learn from mistakes thanks to the ability to feel, understand and act beyond the exclusive individual interests.

### 1.2. COVID-19: Spiritual Intelligence and Social Responsibility

Spiritual intelligence helps determine and characterize the way of living and working in a community, which can be linked to the desire to preserve internal and external peace [51]. This intelligence encompasses the ability to transcend, understand the environment, and engage in what is considered as judicious behavior [52]. It also helps to solve and reflect on everyday problems in search of individual and social well-being [53]. On the other hand, it helps people understand the habitual and domestic practices of human beings in order to give answers to what is lived, and to discover, in these practices, the true meaning of life and values and beliefs emanating from there [54]. This contributes to the construction of a healthy, adaptive system of values, spiritual beliefs, and ethical behaviors, all of which contribute to overcome and sometimes avoid the troubles of life [55]. In this context, spiritual intelligence can help to assimilate the ravages caused by COVID-19, being essential to understand the subjective vitality and the achievement of human wholeness [56].

On the other hand, governments and supranational organizations have promoted measures and some protective and mitigating provisions in order to prevent COVID-19 contagion [57]. Controlling these types of communicable diseases depends mainly on the knowledge, attitudes, practices, and behavior of the population regarding their daily interactions, which underlie socially responsible behavior [58]. One example of this is the use of masks, physical distancing, hand washing, and itinerant confinement [59], setting a real challenge for the individual and society in general. In this scenario, preventive measures can be stimulated with greater force in contexts where comprehensive training has been sought in terms of behavioral awareness, where social responsibility and spiritual intelligence flow [60]. Hopefully, this motivates higher education institutions to install strategies for the creation of habits, respectful treatment, and humanized attitude in interpersonal and intrapersonal relationships [61]. Education thus cannot be biased, but must be able to integrate knowledge related to ethics, social responsibility, and spiritual intelligence, contributing to developing the ethical perspective of care, such as self-care, care of the person, and coexistence among people [18] through a holistic vision, which includes biological, psychosocial, and spiritual well-being. The latter is considered as a perspective that grants a sense of connection with the other senses of life, and with the other abilities of the human being [28].

### 1.3. Spiritual Intelligence and Cutting-Edge Science

It can be pointed out that cutting-edge science is linked to spiritual intelligence, due to the proposals for what these constructs possess in terms of their adaptability and complexity for their scientific study and practical implementation. In this framework, the categories that facilitate its conceptualization are raised: (1) It can be understood as a quantum catalyst, insofar as it enables the holistic and fully integrated functioning of the cerebral cortex through interhemispheric unification, enhancing nonlinear reasoning pathways [62]; (2) It can be approached as an ontological mediator, because it facilitates the achievement of a supra-rationality, i.e., arriving at a vision of reality where the subtle and the manifest are integrated, which facilitates making sense and giving value to previously discordant experiences from the personal and social being [63]; (3) It can be conceived as a chaordic ability, in response to the fact that it aids the personality to operate in the midst of discordant, highly stressful, and volatile scenarios through performances of lateral and divergent nature [64]; (4) It can be considered a metacognitive capacity of necessary approach in higher education in order to transcend the so-called crisis of significance that is occurring in the midst of the industrial, technological, and social revolution [65]; and (5) It can be experienced as a manifestation of inner wisdom and the application of inherited ancestral wisdom in order to achieve full personal and communal development, i.e., more than a discovery, the development of the spiritual dimension is a reencounter with knowledge present in all members of the social fabric [66].

Likewise, within cutting-edge science, it is necessary to highlight the dialogue between spiritual intelligence and neuroscience, particularly informational neuroscience (IN), due to its transdisciplinary and complex nature. IN lays its foundations within Sociobiological Informational Theory (SIT) [67,68], which has seen sustained development within the current scientific context [69,70,71,72]. In the IN vision, spiritual intelligence could be explored from the quantum, cellular, tissue, systemic, paleo-cortical, neocortical, and social levels, opening with these categories of analysis integral enough to engage spiritual intelligence during highly complex social systems. Spiritual intelligence and socially responsible behavior during a pandemic thus contribute to the understanding and installation of habits which correspond with the measures established to stop the spread of the virus, which has its links with cutting-edge science. In this context, having high spiritual intelligence due to the efforts developed by HEIs could motivate the socially responsible behavior that seeks the social welfare of a broader society, thanks to the welfare of the self and its connections with the other, giving importance to the transcendent, leaving behind the superficial, and contributing to social progress.

Finally, studies on social responsibility and spiritual intelligence have considered the research subjects’ sociodemographic characteristics, such as: gender, age, occupation, years of study, territory, education, and income level [24,27,73]. In this case, the aforementioned consideration contributes to the design of strategies, policies, and guidelines for organizations of various natures, which allows for meeting the needs of university students [25,26].

## 2. Methodology

The design of this research is quantitative, exploratory, and sectional. This is done due to the use of two quantitative scales that were applied at one time to university students. The participants were university students from a city located in south-central Chile. A total of 415 participations were collected, of which 362 applications were valid. See Table 1 for the characteristics of the participating subjects.

The research instrument has three sections. The first section is composed of filter questions for the exclusive participation of university students. The second section has questions to help characterize university students: gender, age, year of study, online classes, and number of people at home. The third section has two scales. The first is a spiritual intelligence scale called a spirituality questionnaire (see Table 2), which was designed and validated by Ardiles et al. [74]. The second scale is a unidimensional construct that was extracted from the proposal of Ruiz et al. [75], called a COVID-19 attitudes questionnaire (see Table 3). Both questionnaires’ responses were on a Likert-type scale, where 1 = minimum agreement value and 4 = maximum agreement value. This let us survey the degree of similarity between the statements provided in each of the dimensions and the normal development of the decisions taken during the pandemic, which allowed systematizing attitudes in correspondence with the object of study.

The instrument was applied online between August and October 2021 via Google Forms^®^ due to the health crisis. A link for direct access to the instrument was made available. Each of the applications had a cover sheet where the participants were informed of the objective of the research and anonymity and confidentiality were assured. It was pointed out that participation is voluntary and granting answers did not generate an impact on the students’ health, nor did it generate economic benefits during or after the research. Once the applications were prepared, they were exported through the extension provided by Microsoft Excel. The spreadsheet was managed with the statistical program SPSS version 18.

A matrix analysis was developed through an exploratory factor analysis (EFA), for the arrangement of constructs that facilitated the analysis and approach to the study objective, in order to find the internal structure of the test [76]. Tests were then applied for its verification, using Kaiser–Meyer–Olkin (KMO) and Bartlett’s test of sphericity. The principal component method was used through the Varimax rotated solution, which simplified the model. Clearer results were obtained to identify the factors in each component [76]. Subsequently, internal consistency coefficients were determined through Cronbach’s Alpha [77]. An inferential analysis was developed, for which normality tests were applied to determine the behavior of the data distribution, considering the sociodemographic characteristics of university students, which helped discern the appropriate descriptive statistics and dispersion measures according to the behavior of the data.

## 3. Results

### 3.1. Exploratory Factor Analysis and Descriptive Statistics

The tests developed to verify an adequate exploratory factorial solution were Kaiser–Meyer–Olkin, with a value of 0.933, and Bartlett’s sphericity test, with values of chi2 = 6662.128; gl = 325; and a *p*-value < 0.000. These results allow us to affirm that the factorial matrix is adequate [76]. Regarding the Spirituality Questionnaire, it is composed of the dimension Self-awareness: v1, v2, v2, v3, v4, v5, and v6. The dimension Spiritual practices and beliefs includes the variables: v11, v12, v13, v14, v16, v17, and v18. The final dimension, Spiritual needs, includes the variables: v21, v22, v24, v25, v26, v28, and v29. The variables v8, v15, v18, v20, v23, and v27 were eliminated. The Questionnaire of attitudes towards COVID-19 is composed of constructs that constitute a dimension called Attitudes towards COVID-19, composed of the variables C1, C2, C3, and C4, with variables C5, C6, C7, and v7 being eliminated (Table 3). These explain 65.56% of the data variance. As for the mean, median, standard deviation, and Cronbach’s alpha, the highest value can be identified in the Spiritual needs dimension, due to practices that seek to discover the purpose of life and find inner peace, as well as to foster relationships between people and a meaningful life (mean = 3.67; median = 4; SD = 0.679). On the other hand, the internal consistency analyses in each of the dimensions are high (See Table 4).

### 3.2. Inferential Analysis

For the adequate exploration of statistically significant differences, Kolmogorov–Smirnov and Shapiro–Wilk normality tests were performed to analyze data distribution and homoscedasticity according to each sociodemographic characteristic of the university students. This led to the application of the Kruskal–Wallis and Mann Whitney U non-parametric H tests. Statistically significant differences were only found according to sociodemographic criteria: gender and age criteria. In relation to what was found above, a similar situation appeared in the investigations of Rashidi et al. [29], Becerra and Becerra [31] and Singla et al. [78]. Therefore, the analyses continued considering the sociodemographic characteristics that presented statistically significant differences, the same criterion which was applied by Severino-González et al. [79] and Acuña-Moraga et al. [80]. 

Table 5 shows the mean, median, standard deviation, and *p*-value according to gender. Statistically significant differences are identified in the dimensions Spiritual practices and beliefs (*p*-value = 0.00), Spiritual needs (*p*-value = 0.001), and Attitudes towards COVID-19 (*p*-value = 0.046). 

Regarding the dimension Spiritual practices and beliefs (see Table 5), the highest value is found in the female group, given the search for relationships based on harmony and spiritual balance which lead to the implementation of actions to define personal goals and their links with nature and society (mean = 2.66; median = 3; SD = 0.872). Regarding the aforementioned element, some research on social responsibility also finds a higher value in the group made up of women, because women have a greater social awareness and social responsibility that leads them to the materialization of practices that contribute to society and the environment, which are characterized by empathy, solidarity, and altruism [79,81].

There are also statistical differences in the dimension Spiritual Needs (see Table 5), where women are the ones who evidence the disposition of actions on the part of the students linked to the search for peace, in addition to the respective purposes and meanings of life (mean = 3.76; median = 4; SD = 0.549). In this sense, it is important to point out that spiritual needs are linked to strong emotional ties that can be developed with different people and the search for development and meaning in life, all of which can encourage the consideration of practices related to empathy, well-being, self-efficiency, and prosociality [82,83].

Finally, women also express the highest values in the Attitudes towards COVID-19 dimension (see Table 5), due to the development of practices that show self-care, protection of the family, and compliance with instructions from competent agencies (mean = 2.74; median = 3; SD = 0.478). The COVID-19 pandemic has presented various challenges leading to the deployment of actions, which on one side aim at self-care, and, on the other, the development of socially responsible behavior, responding to new challenges arising in postmodernity and cutting-edge science, due to the constant search for adaptability and social complexity [23,63,64,65].

Table 6 shows the mean, median, standard deviation, and *p*-value according to the students’ age. Statistically significant differences can be observed in the Spiritual needs dimension (*p*-value = 0.003) and in the Attitudes towards COVID-19 dimension (*p*-value = 0.008). 

The highest values in the Spiritual Needs dimension are expressed by the group of students between 18 and 24 years of age (see Table 6), due to the search for the purpose of life, in addition to the necessary inner balance (mean = 3.73; median = 4; SD = 0.604). Spiritual needs are linked to the search for the mysteries of life, the cultivation of interpersonal relationships, emotional beauty, and inner peace, which is found with greater presence in the group of students between 18–24 years of age. All these factors lead to the deployment of efforts that evidence principles and values promoted by social responsibility, which can be propitiated by the understandings linked to cutting-edge science, as a catalyst and descriptor of human behavior [36,66,72]. 

In this sense, statistically significant differences were also found in the dimension Attitudes towards COVID-19 (see Table 6). The highest value also appears among students age 18–24, due to the actions developed that seek self-care, which implies assuming responsibilities that echo the regulations imposed by the government and due to the side-effects of COVID-19 (mean = 2.73; median = 3; SD = 0.497).The group of students between 18–24 years old are the ones who present a higher assessment due to the recognition of actions related to self-care and compliance with regulations seeking to reduce COVID-19 infections. The previous point presents aspects related to social responsibility and cutting-edge science, as a means seeking to understand the personality of people as a consequence of revolutions and crises that have characterized society throughout history [67,79,83].

## 4. Discussion

The current SARS-CoV-2 pandemic scenario has been recognized and characterized for being a hypercomplex process of high re-significance, reenactment, and reimagination of each of the constituent elements of life on a global scale. This scenario must be addressed to the extent of its first impacts on humanity. We are witnessing the initial phases of the virus development within the planet, and the end of the pandemic is not foreseen, at least in the medium or long term. Given this, and considering the multidimensional effects of the pandemic on humanity, it is worthwhile to recognize how human beings are responding to this unprecedented event. Gnatik [84] and Krzyzanowski [85] point out that the new normality merits a sophistication of the ontological and epistemological analysis processes for reality and society. These lead to IST [68], evidencing connections with spiritual intelligence. These elements should encourage the search for the genuine meaning of life in each of the students [86]. This search has relevant actors, including universities, which should promote curricula that install elements of transcendence and common good in the future professionals.

Spiritual intelligence has exhibited significant relationships with vitally important constructs within the pandemic context, including human behavior, cutting-edge science, and IST. In this regard, Hendijani et al. [87] found a positive relationship between spiritual intelligence and social entrepreneurship, which generates a deeper understanding of life, elevated values, a strong sense of purpose, and high motivation levels. To that end, Akhtar et al. [88], through a literature review, concluded that spiritual intelligence training should be included in organizational sustainability training in a broad, convergent, and systemic sense. This leaves room for its approach and applicability due to the low amount of research [89]. Theoretical and practical gaps arise from this situation that need to be addressed in an increasingly liquid and changing society, from which needs related to spirituality emerge.

Avant-garde science seeks to explain the configuration of personalities, which installs understandings and habits that characterize people in their decision-making processes and, therefore, in their attitudes [62,64,68]. These insights allow us to understand some approaches to the needs, practices, and awareness of interpersonal relationships [39,43,54], which favor the development of values linked to social responsibility such as empathy, solidarity, and helping [14]. These provide inputs for the design of institutional strategies that promote socially responsible behavior and contribute to social transformations [90]. When analyzing the results of the instrument used to explore university students’ spiritual intelligence attitudes from the perspective of social responsibility, considering the sociodemographic characteristics of the research subjects during the COVID-19 pandemic, the results indicate that what leads to a developed spiritual intelligence are spiritual needs. These have a significant relationship with the spiritual practices and beliefs of people. In this sense, Caccia and Elgier [91] explain that spiritual needs lead to practices associated with the capacity for self-efficiency and effectiveness, collaborating in psychological well-being, inner peace, and the encounter with oneself in the search for transcendence. These points are also related to spiritual belief, since having spiritual beliefs raises capacity for purpose and meaning of life in the student, given that the factors with the largest impact on the quality of life are those considered as transcendent. The aforementioned provides inputs for the design of educational policies that promote the constitutive values of social responsibility and the principles of spiritual intelligence.

The results reveal the presence of statistically significant differences according to the research subjects’ gender. Considering the surveyed students’ gender and spiritual intelligence results, the results show that, of the total number of participants, women are the ones who mostly have more spiritual beliefs. This leads them to practices that allow them to develop within society with a more comprehensive and humane view in all aspects, allowing them to find greater meaning in life. A study conducted by Rashidi et al. [29] examines the balance of spiritual awareness and related factors in women, where 388 women with different realities were surveyed. In their results, regardless of occupation, responsibility at home, and educational level, women displayed greater spirituality. They argue that this can be explained by the fact that they have a strong awareness through positive thinking to face situations, thus showing a directly significant relationship. This is similar to the study conducted by Seifi et al. [33] in university students. In addition to finding significant relationships between personality dimensions and spiritual intelligence, they concluded that women have higher values of spiritual intelligence, especially because of the extraversion trait. These findings provide tools for decision processes that allow for designing policies and institutional guidelines leading to the installation of aspects related to socially responsible behavior and spiritual practices.

However, the results showed significant differences among the older respondent age group. This shows up particularly with the dimension of spiritual need, since spirituality can vary through the development of the person, depending on their life cycle stage [78]. In a recent study on spiritual intelligence in 474 Peruvian adults during the pandemic, age presented significant differences. These adults were divided into three age groups (young people aged 18–29 years; adults aged 30–59 years; older adults aged 60 years and over). The oldest age group had the highest mean value. This fact lets us infer that, as age advances, people become more conscious in all aspects, which could encourage their participation in social problems and territorial demands [31].

## 5. Conclusions

The work developed with the respective results obtained in this research, and aims to explore university students’ spiritual intelligence attitudes from the perspective of social responsibility considering the sociodemographic characteristics of the research subjects during COVID-19 pandemic, allowing us to infer the following conclusions. (1) For the improvement of future professionals to be comprehensive in all areas, HEIs must strengthen and develop spiritual intelligence, which is the basis of personal development in all its aspects. It is transcendent when developing simultaneously to the other levels of intelligence, which makes it necessary to consider cutting-edge science. (2) Spiritual intelligence is a valuable tool that helps people lead meaningful lives. It empowers students to be aware of who they are, what they do, and how they relate to their fellow humans. This responsible behavior allows them to make decisions with care and awareness, so that the person can conceive the social norms established with compassion and understanding and can develop smoothly within their social environment. (3) A good strategy that should be integrated into universities’ programs is to enhance their students’ spiritual intelligence through avant-garde science. They would achieve a strong hallmark by ensuring that their students develop in this area and face the labor and social world from a holistic, positive, empathetic, and supportive perspective, proposing solutions, transmitting values and knowledge, having responsible awareness of their actions and what they transfer to others; defining the necessary inputs to be extracted from cutting-edge science. (4) Age is an important demographic factor that affects the practices and the spiritual intelligence level of the person. According to the results, the older the person is, the greater the development of this intelligence. This is so because, throughout life, people acquire experiences and learning which serve as a guide for future decisions, being able to reach a higher level of maturity at advanced ages. Considering this, if spiritual intelligence was developed by HEIs from the first formative stages, the student could reach a level of maturity at an earlier age. It is important that educational objectives are rethought within educational centers in order to not only form techno-scientific people, but also, and above all, spiritual and integral citizens. (5) Spiritual intelligence is a discipline of latent possibilities, as it allows the interweaving of such deep and revolutionary fields of study as complex thinking, quantum theory, informational neuroscience, trans-disciplinarity, ancestral wisdom, artificial intelligence, and avant-garde science. (6) It is important that universities develop strategies considering their students’ gender, since this sociodemographic characteristic affects effectiveness when implementing strategies that seek to promote spiritual intelligence and social responsibility, especially since women find expression with stronger opinions and, at the same time, have a greater social conscience that could contribute to the development of socially responsible strategies. (7) It is relevant that higher education institutions implement policies that meet the needs of students according to age group, because experience and maturity have implications for the development of spiritual intelligence and social responsibility as future professionals. (8) It is important to develop research from the philosophical perspective of spiritual intelligence, which contributes to the understanding of spirituality and the philosophical and anthropological particularities of university students. (9) It is transcendental to develop a study on spiritual intelligence based on specific religious formation, since university students’ religious identity in the latter years of their respective majors can affect their attitudes, perceptions, and behavior. (10) Finally, it is hoped that research proposals will be put forward in the future so that, within national policies, university models, or curricular programs, students can be encouraged and motivated to progress in developing spiritual intelligence, understood as an educational meta-skill towards socially responsible and ethically sustainable citizenship.

## Figures and Tables

**Table 1 ijerph-19-11911-t001:** Participants’ characteristics.

Characteristic	Criteria	Percentage (%)
Gender	Male	35.9%
	Female	64.1%
Age	18–24 years	77.6%
	25–31 years	16.4%
	32–38 years	6.0%
School year	First year	34.9%
	Second year	18.3%
	Third year	17.6%
	Fourth year	15.9%
	Fifth year	13.3%
Online classes	Yes	93.7%
	No	6.3%
Number of people cohabitating	Living alone	2.9%
	Living with one person	15.6%
	Living with two people	28.5%
	Living with three or more people	53.0%

**Table 2 ijerph-19-11911-t002:** Spirituality questionnaire.

Dimensions	Items
Self-awareness	V1. I feel satisfied with the person I am
	V2. I have many qualities
	V3. I have a positive attitude toward myself
	V4. I am a valuable person
	V5. I am generally self-confident
	V6. My life has meaning
	V7. I believe that I am the same as others
	V8. I am a compassionate person
	V9. I bring out the positive in difficult situations
	V10. I generally think positive thoughts
Spiritual beliefs	V11. My spirituality helps me to define my goals in life
	V12. My spirituality helps me decide who I am
	V13. My spirituality is part of my overall approach to life
	V14. Spirituality is integrated into my life
Spiritual practices	V15. I am involved in environmental programs
	V16. I read about spirituality and/or self-help
	V17. I do meditation or prayer to achieve inner peace
	V18. I try to live in harmony with nature
	V19. I try to find moments to expand my spirituality
Spiritual needs	V21. I seek a purpose in life
	V22. I enjoy listening to music
	V23. I need to find answers to life’s mysteries
	V24. It is important to me to maintain interpersonal relationships
	V25. I need to achieve inner peace
	V26. I seek the physical, spiritual, and emotional beauty of life
	V27. I need to have a strong emotional bond with people
	V28. My life is evolving
	V29. I need to develop a meaningful life

**Table 3 ijerph-19-11911-t003:** Questionnaire of attitudes towards COVID-19.

Dimensions	Items
Attitudes towards COVID-19	C1. I am interested in self-care and caring for my family.
	C2. I am responsible for my self-care
	C3. I comply with government-mandated rules.
	C4. I prefer to perform self-care, given that there is insufficient availability of health services
	C5. I consider that medicalization solves health problems
	C6. I let the State take care of my health
	C7. Being healthy is a result of the responsibility of my individual self-care

**Table 4 ijerph-19-11911-t004:** Matrix of rotated components of spiritual intelligence and attitudes towards COVID-19; mean, median, standard deviation (SD), and Cronbach’s alpha.

Dimensions				
Variables	Self-Awareness	Spiritual Practices and Beliefs	Spiritual Needs	Attitudes towards COVID-19
v1	0.825			
v2	0.779			
v3	0.836			
v4	0.796			
v5	0.816			
v6	0.751			
v9	0.617			
v10	0.777			
v11		0.823		
v12		0.831		
v14		0.842		
v16		0.686		
v17		0.669		
v19		0.732		
v21			0.699	
v22			0.675	
v24			0.678	
v25			0.653	
v26			0.606	
v28			0.583	
c1				0.746
c2				0.790
c3				0.722
c4				0.747
Mean	2.85	2.49	3.67	2.68
Median	3	3	4	3
SD	0.824	0.926	0.679	0.548
Alfa	0.929	0.919	0.843	0.804

**Table 5 ijerph-19-11911-t005:** Mean, median, and standard deviation (SD) by dimension according to students’ gender.

Gender	Statistics	Self-Awareness	Spiritual Practices and Beliefs	Spiritual Needs	Attitudes towards COVID-19
Male	Mean	2.86	2.20	3.50	2.59
	Median	3	2	4	3
	SD	0.894	0.923	0.842	0.646
Female	Mean	2.84	2.66	3.76	2.74
	Median	3	3	4	3
	SD	0.785	0.872	0.549	0.478
*p* value		0.653	0.00	0.001	0.046

**Table 6 ijerph-19-11911-t006:** Mean, median, and standard deviation (SD) by dimension according to students’ age.

Age	Statistics	Self-Awareness	Spiritual Practices and Beliefs	Spiritual Needs	Attitudes towards COVID-19
18–24	Mean	2.83	2.48	3,73	2.73
	Median	3	3	4	3
	SD	0.795	0.909	0.604	0.497
25–31	Mean	2.89	2.59	3,41	2.56
	Median	3	3	4	3
	SD	0.883	0.962	0.922	0.691
	Mean	2.95	2.43	3.48	2.38
32–38	Median	3	2	4	2
	SD	1.071	0.926	0.750	0.669
*p* value		0.511	0.611	0.003	0.008

## Data Availability

The data presented in this study are available from the corresponding author on reasonable request.

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
