# Peer review of "Social Responsibility and Spiritual Intelligence: University Students’ Attitudes during COVID-19"

_ijerph, 2022, doi:10.3390/ijerph191911911_

Round 1

Reviewer 1 Report

First of all, the author did a very thorough literature review. The topic is very interesting and important. However, the structure and method have big flaws.

Covid is part of the title of the manuscript and is also mentioned in the first sentence of the abstract. So Covid is the most important background information of this study. However, the author did not mention the Covid in the introduction. Readers would not why you want to study this topic under the Covid background in this case.

The research question is to describe university students' spiritual intelligence attitudes during the pandemic. However, the data analysis actually is about gender differences and age differences in spiritual intelligence. For a quantitative study, the authors should say their RQ is whether there is a gender difference in spiritual intelligence. Also, the data is only from students of one city. It is hard to represent the whole university population in Chill or those from other countries. So it is hard to say this is a descriptive study of university students. Authors should modify this. The results did not answer the research question right now. 

Furthermore, the authors reviewed tons of literature and discussed the definition and significance of spiritual intelligence in the literature review part, but they seemed not related to the results. Results are all about gender and age. They are not even mentioned in the literature review. We have no clue why authors examine gender and age. Why not race/ethnicity? why not occupation? why not household income?

Last,the final conclusion and implication do not reflect what authors found in the data. The discussion did not tell us why there are age or gender differences. The implication has nothing to do with gender and age either.

Reviewer 2 Report

The current research utilises two different questionnaires to trace university students’ spiritual intelligence and their attitudes towards COVID-19. However, there are some comments to be considered before publication.

Some keywords, e.g. teaching and educational policy is not adequately discussed in this work. It is suggested to change to other relevant keywords within the spiritual intelligence framework if the system allows.

Introduction:

Critical discussion of the relevancy for COVID-19 to spiritual intelligence should be provided in the early part of the introduction.

The information provided for the avant-garde science is not very relevant to the study and the discussion. Since the idea of spirituality questionnaire is based on a spirituality questionnaire, the author could provide some discussion on the four dimensions instead.

Methodology:

The author stressed the current research as a sectional research. This may not bring much value in explaining the significance of this research method.

The idea of social responsibility might explicitly stated if it is related to the questionnaire of attitudes towards COVID-19.

Since the spirituality questionnaire is validated, the author may have to further explain the rationale of using EFA.

For consideration, the author could further explain the adoption of a 4-point Likert scale.

Results:

The author could provide some discussion to the social responsibility dimensions.

For consideration, the author could provide citation the reason of conducting normality tests.

It is understood that 18-24 appears to be the majority age group for university students. The author may provide further details on the nature of the other mature students and the implications of the score differences.

Discussion:

The earlier parts of the discussion could be further support by the results.

The discussion for the four dimensions of spiritual intelligence is not adequate. It is claimed that “the results indicate that what leads to a developed spiritual intelligence are spiritual needs.” While it is observed that this dimension has a high median (4 out of 4) and significant statistical difference among different genders and age groups, would the author also provide some discussion for dimensions without significant difference (e.g. self awareness) or items with low score (e.g. spiritual practices and beliefs)? The author can also provide some implications on education policy for this dimension with a high median score.

Some policy implications should be provided based on the gender and age differences.

Conclusions:

In general, this research design adequately explains students’ spiritual intelligence and social responsibility during COVID-19 as two main blocks. However, it would be much more interesting how these two are interrelated if more implications on educational policy to be proposed. This would require a change in the methodology and further exploration of the data.

The author could also provide some general overview of Spiritual Intelligence of university students before COVID-19 so that it could better “describe university students’ spiritual intelligence attitudes during the pandemic” by revisiting the spiritual intelligence scores. Perhaps another way of doing so is to explore why some questionnaire items is not useful during the pandemic, given that the questionnaire was validated.

Reviewer 3 Report

The necessity of conducting this study should be clearly stated in the introduction

The discussion and conclusion are not strong enough and be supported by more studies in this area.

It is better to say which inclusion criteria should the students have?
If the correlation between spiritual intelligence and social responsibility had been done and reported, the article would have gained a better position from a scientific point of view.

If possible, the table of demographic characteristics of the participants should also be published (field of study, ....)

Round 2

Reviewer 1 Report

revised version looks good

Author Response

Dear Reviewer 1
Thank you for your review, and for your comments. 
We are pleased that the edited version has satisfied your observations. 
Sincerely, 
Pedro Severino & Co-authors

Reviewer 2 Report

The changes have addressed all my comments.

It is suggested to check the figures in table 1 and others. It is stated that 0% of population is aged 32-38 which does not match table 6.

Author Response

Dear Reviewer 2: 

In response to your observation: 

It is suggested to check the figures in table 1 and others. It is stated that 0% of population is aged 32-38 which does not match table 6.

We have corrected this mistake in Table 1. It was actually 6.0% for ages 32-38 and 0.0% for ages 39+, which we removed from the table. 

Thank you for your detailed review which helped us greatly improve the manuscript. 

Kind Regards
P. Severino + Colleagues